# Behavioral Activation System and Early Life Parental Abuse Are Associated with Antisocial Behaviors in Mexican Adolescents

**DOI:** 10.3390/ijerph19031584

**Published:** 2022-01-30

**Authors:** Jennifer Lizeth Espinoza-Romero, Martha Frías-Armenta, Marc Yancy Lucas, Nadia Sarai Corral-Frías

**Affiliations:** 1Interdisciplinary Graduate Program in Social Sciences, University of Sonora, Hermosillo 83000, Mexico; a209201377@unison.mx; 2Law Department, University of Sonora, Hermosillo 83000, Mexico; 3Psychology Department, University of Sonora, Hermosillo 83000, Mexico; mylucas@email.arizona.edu (M.Y.L.); nadia.corral@unison.mx (N.S.C.-F.)

**Keywords:** behavioral activation system, reinforcement sensitivity theory, early life parental abuse, antisocial behaviors, Mexican adolescents

## Abstract

Antisocial behavior (AB) is a complex phenomenon, predicted by a wide range of biological, environmental, and personality factors. These have high human and economic costs especially in adolescents, highlighting the importance of investigating factors that may be associated with these behaviors. Among the most potent predictors of AB are early life experiences and personality. To this end, the present study sought to investigate the association between early life parental abuse and behavioral activation system (BAS) personality traits assessed within the reinforcement sensitivity theory (RST) framework and antisocial behaviors in Mexican adolescents. Our sample consisted of 342 adolescents (Mage = 17, SD = 2.47) from northwestern Mexico. Participants, after parental consent and participant consent/assent (if minors), self-reported early life parental abuse, current BAS personality traits, and antisocial behaviors. Through structural equation models, our results suggest there is a positive association between early life parental abuse and antisocial behaviors, as well as a negative association with BAS personality traits (R2 = 37%). These results contribute to the current literature by suggesting that personality and environmental variables can predict adolescent antisocial behaviors. Future studies should explore the interplay between these variables longitudinally and investigate both risk and protective factors, as well as negative and positive outcomes.

## 1. Introduction

Antisocial behavior (AB) is a complex phenomenon that can be influenced by a wide range of biological, environmental, and personality factors. AB carries with it high human and economic costs [1] and disproportionally affects adolescents [2]. Of the approximately 30 million crimes reported in Mexico in 2019, a third were carried out by individuals under 25 [3]. These young individuals considered adolescents [4], in many cases, have been coerced by organized crime or by other adults to participate in criminal activities. Early life experiences, particularly those characterized by trauma, abuse, and violence, may predict antisocial behaviors later in life. Likewise, the sociophysical context of the individual may influence behavioral response and exposure to the AB and actions of others. Mexican citizens, most importantly children and adolescents, have experienced a dramatic and unprecedented increase in violent crime starting in 2008, affecting children and adolescents disproportionally in certain segments of the population. According to official data reported by the National Institute of Statistics and Geography [5], homicides in Mexico were stable from the mid-1990s through 2007. However, between 2007 and 2010, the number of reported murders almost tripled. These rates have continued to increase, to the point that in 2020, the rates were almost double those of 2009 (19,000 to 36,000). Conflicts related to the drug trade have had an increased spillover of violence onto civilian non-actors [6], and some of the current highest homicide rates are in northwestern Mexico, where this research was carried out [7]. Despite these high rates, it is well known that increased educational attainment has a bidirectional relationship with crime —those who have a higher education have a lower probability of being involved in crime [8]; however, during the Drug War, this was not the case, and increases in crime have been consistently on the rise ever since. Nevertheless, if abuse, violence, and trauma represent negative factors influencing behavior, then other factors related to approach behaviors such as persistence in the pursuit of achieving personal goals, reinforcement of personal interests, and responsivity to rewarding stimuli may serve as counterbalancing measures in the avoidance of AB. Examining the interplay of these factors and their relationship to negative outcomes can further inform our understanding of the psychosocial foundations of antisocial behaviors. This may be particularly relevant in children and adolescents, given the associations between abuse in childhood and future negative outcomes.

Evidence suggests that antisocial behaviors peaks during adolescence, with most individuals engaging in activities that can be considered antisocial to some degree. For the grand majority, this is a transitional period, and behaviors are not necessarily continued into adulthood [9]. However, AB may persist past adolescence and lead to dire consequences. Across Mexico, approximately 12,000 children/adolescents between the ages of 14 and 18 have been arrested, and of the 10 million nationwide, 7136 have been institutionalized [10,11]. Individual differences in personality are potential predisposing factors that may provide insight into the persistence of these behaviors throughout adolescence and into adulthood.

Identifying the underlying factors that may be associated with AB in adolescents has led to research from a myriad of academic and disciplinary perspectives. One such avenue focuses on theoretically relevant personality traits such as the behavioral approach and inhibition systems (BIS/BAS, respectively) [12]. The study of approach and inhibition is based on the supposition that major personality traits represent basic, individual systems of approach and avoidance. From this perspective, both human and non-human animals are motivated to maximize their exposure to rewards (BAS) and to minimize their exposure to punishment (BIS). A more recent theoretical perspective that developed out of BIS/BAS theory, the reinforcement sensitivity theory (RST) [13], is conceptualized in terms of emotion, motivation, and learning. This theory posits BAS as sensitivity to appetitive stimuli that results in motivated, goal-directed approach behaviors. Given these underlying assumptions, researchers have hypothesized that AB may arise from dysregulation in the reward system [14], and indeed, behavioral studies appear to suggest that individuals engaging in antisocial behaviors may be hypersensitive to rewards [15,16].

The behavioral activation system has been linked to externalizing and antisocial behaviors, as well as reactive and proactive aggression [17,18]. High BAS has been shown to predict drug and alcohol abuse, psychopathy, and hyperactive–impulsive symptoms [19] as well as aggressive behavior [20] in students. There is also evidence that sub-constructs within the BAS framework may represent distinct roles. For instance, high sensitivity to rewards has been associated with alcohol consumption in adolescents [21], risky driving, risky decision making, and risky health behaviors [22] as well as psychopathic traits, conduct problems, and alcohol abuse [23]. Reward-related behaviors have also been shown to predict conflict with adults and socialization with antisocial peers [24].

Some have suggested that the BAS construct can be split into future- and present-oriented behaviors [25]. Studies examining BAS from this perspective have suggested gender differences in reporting where AB was predicted by future-oriented BAS in male participants (planning and persistence toward goals and the pursuit of interests) and present-oriented BAS in female participants (reward reactivity and impulsivity) [26]. Another study reported impulsivity demonstrating predictive utility for AB in female participants and goal-driven persistence in male participants [27]. These investigations suggest an association between BAS and AB, albeit one that is likely complex and dependent upon/interdependent with other factors such as gender or time orientation.

Parental child abuse and the role it can play in development have also been shown to be precipitating factors in future AB. Studies have demonstrated associations between abuse, child maltreatment, and harsh parenting, and antisocial behavior in adulthood [26,27]. An international, cross-cultural study of Asian, European, and North American samples reported a strong relationship between child abuse and AB across contexts [28]. Similarly, a systematic review found child abuse to represent the most constant outcome relationship with antisocial traits [29]. Another longitudinal study found associations between physical abuse and delinquency, externalizing behavioral problems, and drug abuse [30]. Antisocial personality disorder was likewise related to adverse experiences in youth such as child abuse and neglect [31]. An additional longitudinal study examining AB and ecological factors found abusive parenting, peer victimization, and community cohesion to be important predictors of AB in children and adolescents with ADHD [32]. It has thus been repeatedly shown that being the victim of violence, or indeed merely observing violence in childhood, is a potent predictor of AB [33].

Furthermore, evidence suggests that childhood abuse may represent a continuing AB risk factor into adulthood. A study of the relationship between child abuse and antisocial behavior throughout life found higher levels of AB in adults who experienced maltreatment in their childhood [33]. Adverse childhood experiences (abusive parenting, family violence, substance abuse, divorce, incarceration of a family member) have also been associated with recidivism in juvenile offenders [34]. A meta-analysis found that abused adolescents were two times more likely to be involved in antisocial behavior than their counterparts in adulthood [35]. Furthermore, a study of female participants reporting on a wide range of maltreatment found all forms to be associated with antisocial and risky sexual behaviors, depression, and drug/alcohol use [36].

Although not as widely studied within the RST framework, there is some evidence of early life adversity, including child abuse, having an association with approach behaviors, and associated neural markers, specifically, reward-related behaviors [37]. Some recent studies have also found a negative association with self-report reward drive [38]. However, previous studies did not find any associations with any BAS sub-constructs [39]. These mixed results suggest a complex relationship and that these associations may be sub-construct specific.

The social, affective, educational, and formative support received from parents or guardians is critical for positive childhood development. The lack of a safe and supportive environment can lead to emotional and behavioral problems in children [40]. Consequently, healthy parent–child relationships are crucial for appropriate childhood development [41]. In congruence, a longitudinal study of adolescents consistently found that greater reward responsivity was related to higher parenting quality and, in turn, to externalizing and internalizing symptoms [42].

The effects of both child abuse and BAS on antisocial behaviors among adolescents have not been well studied in general, and specifically, little research has been applied to Mexican samples. Previous investigations have tended to focus on factors such as maltreatment, attachment, favorable attitudes toward AB, social bonds, or social concern and its relationship with AB. Given this potential gap, our study examines the effect of both BAS and child abuse on antisocial behavior among Mexican adolescents using a structural equation model. The research seeks to analyze the effect of abusive parenting, as well as the influence of the reinforcement-sensitivity-theory-based BAS [43] on antisocial behaviors. Based on previous evidence, we hypothesize that child abuse and BAS will have a positive effect on antisocial behaviors, and these constructs will, in turn, be associated with each other.

## 2. Method

### 2.1. Participants

The study sample consisted of 342 adolescent residents of three medium-size urban areas in northwestern Mexico (Mage = 17, SD = 2.47), where 126 identified as a cisgender male, 188 as cisgender female, 7 as not cisgender, and 11 chose not to answer. Most participants were undergraduate students (6.7% junior high school, 55% high school, and 32.1% undergraduate students, with 6.1% choosing not to answer). Participants self-reported having low-to-medium socioeconomic status (SES) [44].

### 2.2. Procedure

Parental consent was obtained prior to first contact with students who were then provided with basic information about the study. Following the consent process, students were provided with a link to the questionnaire via the Qualtrics software package (licensed with the University of Sonora). The informed consent process was approved by the Ethics Committee for the University of Sonora (ID. USO317007147). Questionnaires typically took around one hour to complete.

### 2.3. Measures

#### 2.3.1. Antisocial Behavior

The antisocial behavior scale [45] for Mexican samples was used to measure the frequency of AB for adolescents during the previous 12 months. The instrument is comprised of four scales: vandalism (including behaviors such as smashing bottles in the street, on school grounds, or other places), assault (e.g., involved in gang fights or other gang activities), theft (e.g., stole or tried to steal a motor vehicle), and general deviance (e.g., dishonest behaviors such as littering or not stopping at a stop sign). A total of 21 items were answered through a scale-point scoring system of frequency (ranging from 1 = never to 6 = more than twenty times over the previous 12 months). Previous research has demonstrated acceptable internal consistency for this instrument (Cronbach’s α = 0.75) [46].

#### 2.3.2. Abusive Parenting

The Parent–Child Conflict Tactics Scale was used to measure the degree of parental abuse suffered by the student [47]. The instrument measures the frequency of the parental physical, severe, and psychological abuse, using 18 items including statements such as “She/he threatens you”, “She/he spanks you on the bottom with their bare hand”. Responses were scored as 0 = never, 1= one, 2 = twice, 3 = 3–5 times, 4 = 6–10 times, 5 = 11+. Previous research has demonstrated excellent internal consistency (Cronbach’s α = 0.90) [48].

#### 2.3.3. Behavioral Approach System (BAS)

The Behavioral Approach Scale (BAS) of the Reinforcement Sensitivity Theory of Personality Questionnaire (RST-PQ [49]) for a Mexican population was used to examine the individual behavioral approach. The scale comprised of four related but separate subscales: reward interest (RST-PQ RI), goal-driven persistence (RST-PQ GDP), reward reactivity (RST-PQ RR), and impulsivity (RST-PQ I). A total of 35 items with response options ranging from 1= totally disagree to 4 = totally agree. Previous studies have reported mixed results for internal consistency: Cronbach’s α = 0.63 (RST-PQ RR), 0.77 (RST-PQ GDP), 0.70 (RST-PQ RI) 0.61 (RST-PQ I) [50].

### 2.4. Statistical Analyses

The internal consistency of the measures was calculated using SPSS v.21 software (IBM Mexico, Mexico City, Mexico) (Cronbach´s α) and the results of normality using the Kolmogorov–Smirnov (K–S) test. Confirmatory factor analysis (CFA) was used to form parental abuse, behavioral approach system (BAS), and antisocial behavior factors. Following this, we tested a structural equation model to examine parenting abuse and BAS factors as independent variables of antisocial behavior. Both models were analyzed using EQS [51]. Given that the Mardia multivariate normalized coefficient showed a non-normal distribution of variables, we selected robust methods to test the model. We used the Satorra–Bentler chi-squared test for robust data [52], and BBNNFI, CFI (>0.90), and RMSEA (<0.08) as statistical fit indexes [53]. Given previous gender differences [25], we conducted exploratory mean difference analyses to test if these were present in our data. Normality test (K–S) showed that the data were not normally distributed (*p* < 0.05); thus, non-parametric tests were used (Table 1). Only differences between cisgender males and females were tested due to the low report of non-cisgender participants.

## 3. Results

The descriptive data of the scales are presented in Table 1.

Participants reported low levels of antisocial behavior across the sample. Table 2 contains the rate of antisocial behavior (expressed as a percentage) self-reported by the participants.

The structural equation model confirmed our hypotheses by showing coherence between theoretical specified factors (Figure 1. The child abuse factor was formed by two variables: abuse from mother (λ = 0.52) and abuse from father (λ = 0.84). The behavioral approach system (BAS) consisted of reward interest (λ = 0.66) (RST-PQ RI), goal-driven persistence (RST-PQ GDP) (λ = 0.86), reward reactivity (RST-PQ RR) (λ = 0.69), and impulsivity (RST-PQ I) (λ = 0.05). The antisocial behavior factor (AB) comprised vandalism (λ = 0.86), general deviance (λ = 0.88), theft (λ = 0.89), and assault (λ = 0.79). The structural equation model demonstrated (1) the direct and negative effect of the BAS (structural coefficient, −0.22) and (2) the direct and positive effect of parental abuse on antisocial behavior (structural coefficient, 0.52). Child abuse also demonstrated a negative covariance on BAS (structural coefficient, −0.25). The model demonstrated acceptable goodness of fit (Mardia = 285.11 SBX2 (32 D.F.) = 38.27; *p* = 0.20; BBNFI = 0.96; BBNNFI = 0.97; CFI = 0.99; RMSEA = 0.02), suggesting that the theoretical model adjusted to the data. The model explained 37% of the variance in antisocial behaviors (R^2^ = 0.37).

The exploratory analysis also showed that there were gender differences between cis-gender male and female participants. Briefly, cisgender male participants reported committing more antisocial behaviors, specifically, vandalism and general deviant behaviors. On the other hand, cisgender female participants reported higher parental abuse from their mothers and higher reinforcement interest and goal-driven persistent behaviors (Table 3).

## 4. Discussion

Adolescence can be characterized as a time of increased risk-taking and sensation-seeking behaviors, which, in some cases, can lead to an increased probability of antisocial behaviors [54]. These behaviors carry with them potentially serious and significant human and economic costs, which makes examining the precipitating factors that may lead to AB of critical importance [1]. The present study is one of the first to provide evidence of an association between BAS and childhood abuse and antisocial behaviors in adolescents, especially in an understudied global south sample.

Extant evidence has demonstrated that childhood abuse and harsh parenting may lead to an increased probability of involvement in delinquent behavior [55]. In congruence, our results find that both mother and father maltreatment was associated with increased reports of antisocial behaviors during adolescence. Theoretical work has suggested that parents are the first environment for a child and that negative or stressful events can have a significant effect throughout life [34]. Empirical work has demonstrated that abuse during childhood may have lasting effects that can lead to antisocial behaviors [29,56] especially in male adolescents [57].

Further, our results demonstrated an association between childhood abuse and self-report BAS. This has not been extensively studied, and results have been mixed. Some have found no association [39], while others, in congruence with our results, found a negative association [38]. However, unlike our results, the previous research reported only an association with reward-driven behavior. These differences across studies may be the result of variations in sub-construct responses within the BAS construct. For instance, studies have reported negative associations between reward sensitivity and reactivity and childhood trauma or early life stress [58,59], while others have demonstrated positive associations between adverse early life experiences and increased impulsivity [60,61].

As previously mentioned, our results are congruent with antecedent research demonstrating an association between antisocial behaviors and BAS. However, it is important to point out that, unlike previous studies in the literature, and contrary to our hypothesis, we found a negative relationship with antisocial behaviors. This may be due to impulsivity not loading as strongly with other BAS constructs. Previous research has shown that goal-driven persistence and impulsivity, had a near-zero correlation, suggesting that impulsivity is not as associated with other BAS constructs in other samples [62,63,64]. Constructs within BAS may thus have a conflicting association, in which impulsivity may be a risk factor, and goal-directed behaviors may be protective. Our results are consistent with previous work demonstrating “future” versus “now” traits may have opposing effects [65]. This would likewise be congruent with recent theoretical models, such as the dual-system model, that suggest increased risk taking may be due to an imbalance between the individual’s rapidly developing socioemotional system and their gradually developing cognitive control system [66]. Similarly, research has posited that adolescents are inclined to take risks because they are more sensitive to the rewards and possess an undeveloped or underdeveloped ability to control their impulses [67].

This study does have several limitations. The scale that assessed childhood abuse only inquired about mother and father abuse. Given changes in the makeup of families, it is important to update how we assess early life abuse or trauma. Further, we used a sample that only included participants from three medium Mexican cities from the northwestern region of the country. The study should be replicated in other samples to test generalizability. Further, the self-report of antisocial behaviors was skewed, as only a few participants reported more serious antisocial behaviors. Despite low reports of antisocial behaviors, it is important to study general populations. Therefore, a future avenue of the study could focus on measuring AB on a broad spectrum, ranging from less to more serious infractions to probe the range of variability across the population.

Limitations notwithstanding, the results of our research present evidence that BAS and childhood abuse influence antisocial behaviors in a Mexican adolescent sample. The findings provide replication of previous research from the context of the underrepresented global south. Researchers have emphasized the necessity for psychological science to be more representative of the global human population [68,69], and there have been increasing calls for greater diversity in the psychological sciences [70]. Given the high costs of antisocial behaviors, it is crucial to study the generalizability of results in diverse samples throughout the world.

Previous literature has highlighted the crucial importance of investigating the psychological factors that protect individuals against negative stimuli. Recent research has begun to target both environmental and personality variables, which may be protective of the appearance of antisocial behaviors [71,72]. In this study, we found that some BAS constructs may serve such a protective function. Future research should include risk and protective factors, as well as negative and positive outcomes associated with BAS.

Examining both risk and protective outcomes can inform the development of possible interventions and preventative efforts. Our results suggest the implementation of the positive aspects of the BAS system can lead to decreased antisocial behaviors. Further, recent evidence has demonstrated that risk-taking behaviors, which have been linked to the BAS system, can also lead to positive outcomes in the right context. While much of the previous literature has focused on risk taking as it relates to negative outcomes such as illegal or dangerous behaviors, risks can also be directed toward positive, socially acceptable, and constructive behaviors such as studying abroad or learning a new skill. Despite the possible positive outcomes, little is known about the nature of positive risk taking given the tendency toward negative outcomes, such as substance use or delinquency [67]. Orienting future research toward examining both risk and protective factors, as well as positive and negative outcomes, can uncover important underlying mechanisms associated with antisocial behavior and better inform efforts and promote those factors across contexts.

## 5. Conclusions

Antisocial behaviors carry with them potentially serious and significant human and economic costs, highlighting the importance of examining precipitating factors. The present study provides evidence of an association between BAS and childhood abuse and antisocial behaviors in adolescents in an understudied global south sample. This research is significant because it provides evidence of generalizability of previous results, but also because it extends our knowledge of risk and protective factors for antisocial behaviors. Future work should focus on more diverse samples and longitudinal research design. Finally, to better inform preventative efforts and intervention, it is important to assess risk and protective factors, as well as negative and positive outcomes.

## Figures and Tables

**Figure 1 ijerph-19-01584-f001:**
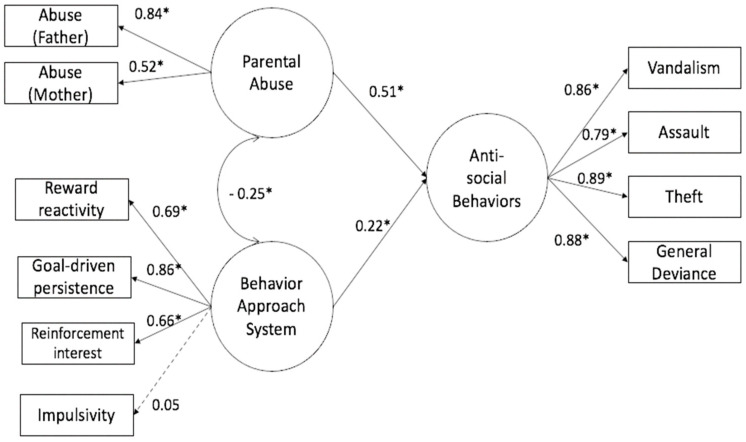
Structural model of child abuse and the behavior approach system and its influence on adolescent antisocial behavior. Mardia = 285.11, SBX2 (32 D.F.) = 38.27, *p* = 0.20, BBNNFI = 0.96; CFI = 0.97; RMSEA = 0.02, R2 = 0.37 * *p* < 0.05.

**Table 1 ijerph-19-01584-t001:** Descriptive Statistics of the scales.

	Mean	SD	K-S (Sig)	Cronbach A
AB				
Vandalism	1.71	0.75	<0.001	0.77
Assault	1.42	1.13	<0.001	0.78
Theft	1.42	1.13	<0.001	0.78
General deviance	1.42	0.78	<0.001	0.86
**Parental** **Abuse**				
Mother	1.28	0.75	<0.001	0.90
Father	1.57	0.97	<0.001	0.87
**BAS**				
Reinforcement interest	3.71	0.48	<0.001	0.90
Goal-driven persistence	3.14	0.89	<0.001	0.91
Reward reactivity	2.71	1.11	<0.001	0.86
Impulsivity	2.71	0.75	<0.001	0.83

Note: K-S = Kolmogorov-Smirnov.

**Table 2 ijerph-19-01584-t002:** Rates of antisocial behaviors in an adolescent sample.

	Vandalism	General Deviance	Thief	Assault
Men	60%	52%	75%	80%
Women	40%	44%	25%	20%

Rates of antisocial behaviors among participants who identified as cis gender male and cis gender female (given that the sample only included seven respondents who identified as other than cis gender, making analysis difficult.

**Table 3 ijerph-19-01584-t003:** Gender differences between variables.

	Gender				
	Female *n* = 188	Male *n* = 127			
	ave. rank	ave. rank	Z	*p*	d
Antisocial Behaviors					
Vandalism	152.84	164.46 *	−2.32	0.02	0.22
Assault	152.68	164.69 *	−2.85	0.004	0.26
Theft	155.32	160.76	−1.36	0.17	-
General deviance	151.43	166.56 *	−2.58	0.01	0.26
Parental Abuse					
Father (parental abuse)	159.04	155.20	−0.784	0.43	-
Mother (parental abuse)	162.18 *	150.18	−2.22	0.03	0.22
BAS					
Reward reactivity	157.05	157.80	−0.079	0.93	-
Goal-driven persistence	164.90 *	146.46	−1.96	0.05	0.17
Reinforcement interest	168.53 *	141.04	−3.04	0.002	0.17
Impulsivity	155.70	160.18	−0.450	0.65	-

Note: * significant difference.

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
