# Peer review of "Behavioral Activation System and Early Life Parental Abuse Are Associated with Antisocial Behaviors in Mexican Adolescents"

_ijerph, 2022, doi:10.3390/ijerph19031584_

Round 1

Reviewer 1 Report

Thank you for the opportunity to review this paper.

Title: Behavioral Activation System and Early-Life Parental Abuse are Associated with Antisocial Behaviors in Mexican Adolescents

The aim of the study is quite interesting. However, the paper has minor limitations that should be addressed:

Lines 32-34, it is stated in the introduction section “Early life experiences, particularly those characterized by trauma, abuse, and violence, may predict antisocial behaviors later in life. Likewise, the sociophysical context of the individual may influence behavioral response and exposure to the AB and actions of others.”

Comment: In the introduction and the discussion sections it should explicitly consider very particular aspects of the social situation in Mexico. The crime figures in Mexico are unfortunately different from those in any other part of the world. It should be considered that Mexico is one of the most crime-ridden countries in the world, and that some of the northwestern states of Mexico, such as Sonora, Baja California, and Sinaloa, are among the most crime-ridden states in Mexico. This reality must be reflected in a study on antisocial behaviour carried out on such a specific sample of the population.

Lines 31-33: “Of the approximately 30 million crimes reported in Mexico in 2019, a third were carried out by individuals under 25 [3].”

Comment: Considering that the under-25s make up a third of Mexico's population, this does not seem very relevant.

Line 137, “The study sample consisted of 342 adolescents residents of northwestern Mexico”.

Comment: this should be more specific. Is the setting urban, rural, did participants come from one locality or several?

Lines 137-138: (Mage= 17, SD= 2.47)

Comment: Mage?

Lines174-176: “Previous studies have reported mixed 174 results for internal consistency: Cronbach´s α = .63 (RST-PQ RR), .77 (RST-PQ GDP), .70 175 (RST-PQ RI) y.61 (RST-PQ I) [47].”

Comment: they are really low internal consistency data.

Regards

Author Response

Review 1

We greatly appreciate these comments and suggestions. We believe they have improved our manuscript. We have provided detailed point-by-point responses to your comments. We hope to have addressed your concerns.

Lines 32-34, it is stated in the introduction section “Early life experiences, particularly those characterized by trauma, abuse, and violence, may predict antisocial behaviors later in life. Likewise, the sociophysical context of the individual may influence behavioral response and exposure to the AB and actions of others.”

Comment: In the introduction and the discussion sections it should explicitly consider very particular aspects of the social situation in Mexico. The crime figures in Mexico are unfortunately different from those in any other part of the world. It should be considered that Mexico is one of the most crime-ridden countries in the world, and that some of the northwestern states of Mexico, such as Sonora, Baja California, and Sinaloa, are among the most crime-ridden states in Mexico. This reality must be reflected in a study on antisocial behaviour carried out on such a specific sample of the population.

Answer: Although it is clear that Mexican citizens, most importantly children and adolescents, have experienced a dramatic and unprecedented increase in violent crime starting in 2008, these affect children disproportionally in certain segments of the population. Our sample includes mostly high school and undergraduate students who are less probable to be exposed to such experiences. We have added some information in the introduction to reflect the situation in northwest Mexico and also clarified that it does not affect all segments. Below is what we have added to the text. Page 1 Lines 38-51:

Mexican citizens, most importantly children and adolescents, have experienced a dramatic and unprecedented increase in violent crime starting 2008, affecting children and adolescents disproportionally in certain segments of the population. According to official data reported by the National Institute of Statistics and Geography (INEGI, 2021), homicides in Mexico were stable from the mid-1990s through 2007. However, between 2007 and 2010, the number of reported murders almost tripled. These rates have continued to increase wherein in 2020 the rates were almost double from 2009 (19,000 to 36,000). Conflict-related to the drug trade has had increased spillover of violence onto civilian non-actors (Brown, 2018) and some of the current highest homicide rates are in northwest Mexico, where this research took place (ENVIPE, 2020). Despite these high rates, it is well known that increased educational attainment has a bidirectional relationship with crime (Gleditsch et al., 2021), where those who have a higher education have less probability of being involved in crime (however during the Drug War this has not been the case where increases in crime have been consistently on the rise).

Lines 31-33: “Of the approximately 30 million crimes reported in Mexico in 2019, a third were carried out by individuals under 25 [3].”

Comment: Considering that the under-25s make up a third of Mexico's population, this does not seem very relevant.

Answer: Thank you for your comment. We included this statistic given that those under 25 are considered by current standards adolescents (Sawyer et al., 2018). Further many of these adolescents are coerced into participating in crimes. The UN convention of children's rights, which establishes that children and adolescents should be protected from any kind of abuse or exploitation (Article, 36, 38). Some children and adolescents are exploited by organized crime or sometimes by other adults. Below is what we have added to the text, Page 1 Lines 33-35:

“These young individuals, considered adolescents (Sawyer et al., 2018), in many cases, have been coerced by organized crime or by other adults to participate in criminal activities”.

Line 137, “The study sample consisted of 342 adolescents’ residents of northwestern Mexico”.

Comment: this should be more specific. Is the setting urban, rural, did participants come from one locality or several?

Answer: Thank you for these comments. The sample is from four medium-size cities of the Sonora state.  We have added this information in the participants' section in the methods. Below is what we have added to the text (bold), Page 4 Line 152:

“The study sample consisted of 342 adolescent residents of three medium-size urban areas in northwestern Mexico.”

Lines 137-138: (Mage= 17, SD= 2.47)

Comment: Mage?

Answer: Thank you for this comment M stands for “mean” and represents the mean of the age.

Lines174-176: “Previous studies have reported mixed 174 results for internal consistency: Cronbach´s α = .63 (RST-PQ RR), .77 (RST-PQ GDP), .70 175 (RST-PQ RI) y.61 (RST-PQ I) [47].”

Comment: they are really low internal consistency data.

Answer: We agree that some of the internal consistency, measured through Cronbach´s, is low (α=.61). However, according to Nunally and Bernstein, (1994) this is still acceptable.

Reviewer 2 Report

Overview:  The language used for different psychological theories should not be mixed without a clear definition - i.e.  externalisation, personality vs teperament;

Some parts of the manuscript are written in difficult to understand form, use of language could flow more smoothly (and it needs proof-reading throughout)

At the same time this article sheds some light on very important topic and broadens the perspective form Mexico sample.

Abstract:  Thorough and concise with good explanation of population, purpose, and findings

Materials and methods: nicely done

Discussion: minor misteakes

272 - it IS important

273 - of THE study

Conclusion: good for not only the results, but also supporting the use of the methodology.

Refernces:

67 - it is an editorial and you cite one of your articles from the same edition of this journal - please state why this double cite or delete this one

Author Response

Reviewer 2

Overview:  The language used for different psychological theories should not be mixed without a clear definition - i.e.  externalisation, personality vs teperament;

Answer: We greatly appreciate your comment. We have looked through the manuscript and hope to have increased the consistency in terminology.

Some parts of the manuscript are written in difficult to understand form, use of language could flow more smoothly (and it needs proof-reading throughout). At the same time this article sheds some light on very important topic and broadens the perspective form Mexico sample.

Answer: We greatly appreciate these comments and suggestions. We have proofread the manuscript and hope to have addressed the readability issue. We have additionally provided a detailed point-by-point response to your additional comments below. We hope to have addressed your concerns.

Abstract:  Thorough and concise with good explanation of population, purpose, and findings

Answer: Thank you for your positive feedback.

Materials and methods: nicely done

Answer: We appreciate your positive comments.

Discussion: minor mistakes

272 - it IS important

Answer: Thank you for pointing this typo out. We have fixed it.

273 - of THE study

Answer: Thank you for pointing this typo out. We have fixed it.

Conclusion: good for not only the results, but also supporting the use of the methodology.

Answer: We appreciate your positive comments.

References:

67 - it is an editorial and you cite one of your articles from the same edition of this journal - please state why this double cite or delete this one

Answer: We appreciate your comment and we understand it may be confusing. Although they have pretty similar topics, one is an editorial and the other is a paper.
